# *U-NO*: U-shaped Neural Operators

**Md Ashiqur Rahman**                                          *rahman79@purdue.edu*
*Department of Computer Science*
*Purdue University*

**Zachary E. Ross**                                           *zross@caltech.edu*
*Seismological Laboratory*
*California Institute of Technology*

**Kamyar Azizzadenesheli**                                    *kamyara@nvidia.com*
*NVIDIA Corporation*

**Reviewed on OpenReview:** *https://openreview.net/forum?id=j3oQF9coJd*

## Abstract

Neural operators generalize classical neural networks to maps between infinite-dimensional spaces, e.g., function spaces. Prior works on neural operators proposed a series of novel methods to learn such maps and demonstrated unprecedented success in learning solution operators of partial differential equations. Due to their close proximity to fully connected architectures, these models mainly suffer from high memory usage and are generally limited to shallow deep learning models. In this paper, we propose U-shaped Neural Operator (*U-NO*), a U-shaped memory enhanced architecture that allows for deeper neural operators. *U-NO*s exploit the problem structures in function predictions and demonstrate fast training, data efficiency, and robustness with respect to hyperparameters choices. We study the performance of *U-NO* on PDE benchmarks, namely, Darcy's flow law and the Navier-Stokes equations. We show that *U-NO* results in an average of 26% and 44% prediction improvement on Darcy's flow and turbulent Navier-Stokes equations, respectively, over the state of art. On Navier-Stokes 3D spatiotemporal operator learning task, we show *U-NO* provides 37% improvement over the state of the art methods.

*Keywords: Neural Operators, Partial Differential Equations*

## 1 Introduction

Conventional deep learning research is mainly dominated by neural networks that allow for learning maps between finite dimensional spaces. Such developments have resulted in significant advancements in many practical settings, from vision and nature language processing, to robotics and recommendation systems (Levine et al., 2016; Brown et al., 2020; Simonyan & Zisserman, 2014). Recently, in the field of scientific computing, deep neural networks have been used to great success, particularly in simulating physical systems following differential equations (Mishra & Molinaro, 2020; Brigham, 1988; Chandler & Kerswell, 2013; Long et al., 2015; Chen et al., 2019; Raissi & Karniadakis, 2018). Many real-world problems, however, require learning maps between infinite dimensional functions spaces (Evans, 2010). Neural operators are generalizations of neural networks and allow to learn maps between infinite dimensional spaces, including function spaces (Li et al., 2020b). Neural operators are universal approximators of operators and have shown numerous applications in tackling problems in partial differential equations (Kovachki et al., 2021b).

Neural operators, in an abstract form, consist of a sequence of linear integral operators, each followed by a nonlinear point-wise operator. Various neural operator architectures have been proposed, most of which focus on the approximation of the Nyström type of the inner linear integral operator of neural operators (Li

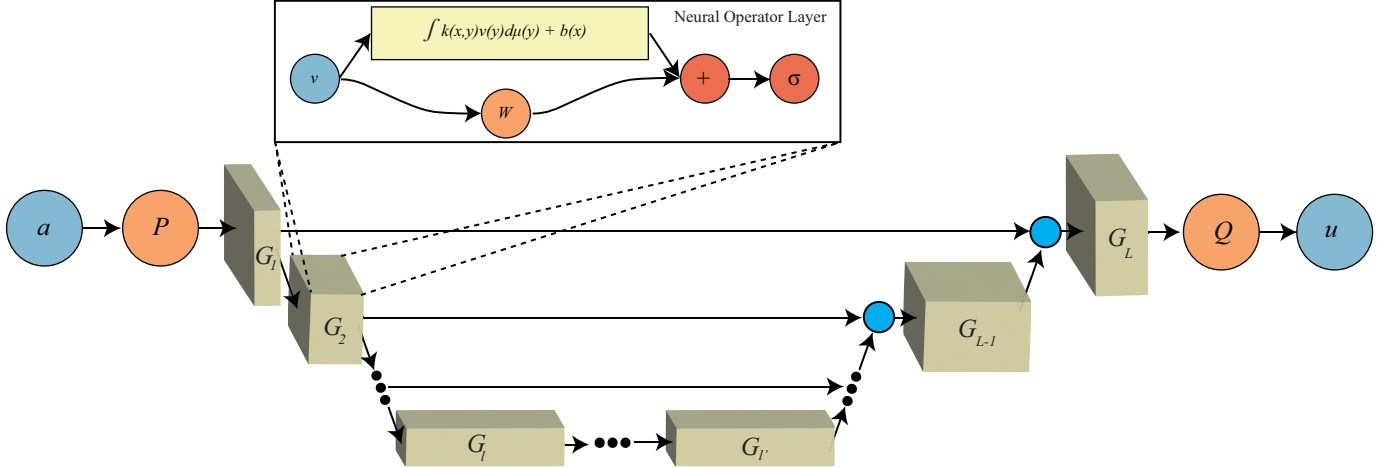

Figure 1: *U-NO* architecture. $a$ is an input function, $u$ is the output. Orange circles are point-wise operators, rectangles denote general operators, and smaller blue circles denote concatenations in function spaces.

et al., 2020a;c; Gupta et al., 2021; Tripura & Chakraborty, 2023). For instance, Li et al. (2020a) suggests to use convolution kernel integration and proposes Fourier neural operator (*FNO*), a model that computes the linear integral operation in the Fourier domain. The linear integral operation can also be approximated using discrete wavelet transform (Gupta et al., 2021; Tripura & Chakraborty, 2023). But *FNO* approximates the integration using the fast Fourier transform method (Brigham, 1988) and enjoys desirable approximation theoretic guarantee for continuous operator (Kovachki et al., 2021a). The efficient use of the built-in fast Fourier transform makes the *FNO* approach among the fastest neural operator architectures. Tran et al. (2021) proposes slight modification to the integral operator of *FNO* by factorizing the multidimensional Fourier transform along each dimension. These developments have shown successes in tackling problems in seismology and geophysics, modeling the so-called Digital Twin Earth, and modeling fluid flow in carbon capture and storage experiments (Pathak et al., 2022; Yang et al., 2021; Wen et al., 2021; Shui et al., 2020; Li et al., 2022). These are crucial components in dealing with climate change and natural hazards. Many of the initial neural operator architectures are inspired by fully-connected neural networks and result in models with high memory demand that, despite the successes of deep neural networks, prohibit very deep neural operators. Though there are considerable efforts in finding more effective integral operators, works investigating suitable architectures of Neural operators are scarce.

To alleviate this problem, we propose the U-shaped Neural Operator (*U-NO*) by devising integral operators that map between functions over different domains. We provide a rigorous mathematical formulation to adapt the U-net architecture for neural networks for finite-dimensional spaces to neural operators mapping between function spaces (Ronneberger et al., 2015). Following the u-shaped architecture, the *U-NO-* first progressively maps the input function to functions defined with smaller domains (encoding) and later reverses this operation to generate an appropriate output function (decoding) with skip connections from the encoder part (see Fig. 1). This efficient contraction and subsequent expansion of domains allow for designing the over-parametrized and memory-efficient model, which regular architectures cannot achieve.

Over the first half (encoding) of the *U-NO* layers, we map each input function space to vector-valued function spaces with steadily shrunken domains over layers, all the while increasing the dimension of the co-domains, i.e., the output space of each function. Throughout the second half (decoding) of *U-NO* layers, we gradually expand the domain of each intermediate function space while reducing the dimension of the output function co-domains and employ skip connections from the first half to construct vector-valued functions with larger co-dimension. The gradual contraction of the domain of the input function space in each layer allows for the encoding of function spaces with smaller domains. And the skip connections allow for the direct passage of information between domain sizes bypassing the bottleneck of the first half, i.e., the functions with the smallest domain (Long et al., 2015). The *U-NO* architecture can be adapted to the existing operator

learning techniques (Gupta et al., 2021; Tripura & Chakraborty, 2023; Li et al., 2020a; Tran et al., 2021) to achieve a more effective model. In this work, for the implementation of the inner integral operator of *U-NO*, we adopt the Fourier transform-based integration method developed in *FNO*. As *U-NO* contracts the domain of each input function at the encoder layers, we progressively define functions over smaller domains. At a fixed sampling rate (resolution of discretization of the domain), this requires progressively fewer data points to represent the functions. This makes the *U-NO* a memory-efficient architecture, allowing for deeper and highly parameterized neural operators compared with prior works. It is known that over-parameterization is one of the main essential components of architecture design in deep learning (He et al., 2016), crucial for performance (Belkin et al., 2019), and optimization (Du et al., 2019). Also recent works show that the model overparameterization improves generalization performance (Neyshabur et al., 2017; Zhang et al., 2021; Dar et al., 2021). The *U-NO* architecture allows efficient training of overparameterized model with smaller memory footprint and opens neural operator learning to deeper and highly parameterized methods.

We establish our empirical study on Darcy's flow and the Navier-Stokes equations, two PDEs which have served as benchmarks for the study of neural operator models. We compare the performance of *U-NO* against the state of art *FNO*. We empirically show that the advanced structure of *U-NO* allows for much deeper neural operator models with smaller memory usage. We demonstrate that *U-NO* achieves average performance improvements of **26%** on high resolution simulations of Darcy's flow equation, and **44%** on Navier-Stokes equation, with the best improvement of **51%**. On Navier-Stokes 3D spatio-temporal operator learning task, for which the input functions are defined on the $3D$ spatio-temporal domain, we show *U-NO* provides **37%** improvement over the state of art *FNO* model. Also, the *U-NO* outperforms the baseline *FNO* on zero-shot super resolution experiment.

It is important to note that *U-NO* is the first neural operator trained for mapping from function spaces with 3D domains. Prior studies on 3D domains either transform the input function space with the 3D spatio-temporal domain to another auxiliary function space with 2D domain for the training purposes, resulting in an operator that is time discretization dependent or modifies the input function by extending the domain and co-domain (Li et al., 2020a). We further show that *U-NO* allows for much deeper models $(3\times)$ with far more parameters $(25\times)$, while still providing some performance improvement (Appendix A.2). Such a model can be trained with only a few thousand data points, emphasizing the data efficiency of problem-specific neural operator architecture design.

Moreover, we empirically study the sensitivity of *U-NO* against hyperparameters (Appendix A.2) where we demonstrate the superiority of this model against *FNO*. We observe that *U-NO* is much faster to train (Appendix A.6) and also much easier to tune. Furthermore, for *U-NO* as a model that gradually contracts (expands) the domain (co-domain) of function spaces, we study its variant which applies these transformations more aggressively. A detailed empirical analysis of such variant of *U-NO* shows that the architecture is quite robust with respect to domain/co-domain transformation hyperparameters.

## 2 Neural Operator Learning

Let $\mathcal{A}$ and $\mathcal{U}$ denote input and output function spaces, such that, for any vector valued function $a \in \mathcal{A}$, $a : \mathcal{D}_\mathcal{A} \to \mathbb{R}^{d_\mathcal{A}}$, with $\mathcal{D}_\mathcal{A} \subset \mathbb{R}^d$ and for any vector valued function $u \in \mathcal{U}$, $u : \mathcal{D}_\mathcal{A} \to \mathbb{R}^{d_\mathcal{U}}$, with $\mathcal{D}_\mathcal{U} \subset \mathbb{R}^d$. Given a set of $N$ data points, $\{(a_j, u_j)\}_{j=1}^N$, we aim to train a neural operator $\mathcal{G}_\theta : \mathcal{A} \to \mathcal{U}$, parameterized by $\theta$, to learn the underlying map from $a$ to $u$. Neural operators, in an abstract sense, are mainly constructed using a sequence of non-linear operators $G_i$ that are linear integral operators followed by point-wise non-linearity, i.e., for an input function $v_i$ at the $i$th layer, $G_i : v_i \to v_{i+1}$ is computed as follows,

$$G_i v_i(x) := \sigma\big(\int \kappa_i(x,y)v_i(y)d\mu_i(y) + W_i v_i(x)\big) \tag{1}$$

Here, $\kappa_i$ is a kernel function that acts, together with the integral, as a global linear operator with measure $\mu_i$, $W_i$ is a matrix that acts as a point-wise operator, and $\sigma$ is the point-wise non-linearity. One can explicitly add a bias function $b_i$ separately. Together, these operations constitute $G_i$, i.e., a nonlinear operator between a function space of $d_i$ dimensional vector-valued functions with the domain $\mathcal{D}_i \subset \mathbb{R}^d$ and a function space of $d_{i+1}$ dimensional vector-valued functions with the domain $\mathcal{D}_{i+1} \subset \mathbb{R}^d$. Neural operators often times are

accompanied with a point-wise operator $P$ that constitute the first block of the neural operator and mainly serves as a lifting operator directly applied to input function, and a point-wise operator $Q$, for the last block, that is mainly purposed as a projection operator to the output function space.

Previous deployments of neural operators have studied settings in which the domains and co-domains of function spaces are preserved throughout the layers, i.e., $\mathcal{D} = \mathcal{D}_i = \mathcal{D}_{i+1}$, and $d' = d_i = d_{i+1}$, for all $i$. These models, e.g. *FNO*, impose that each layer is a map between functions spaces with identical domain and co-domain spaces. Therefore, at any layer $i$, the input function is $\nu_i : \mathcal{D} \to \mathbb{R}^{d'}$ and the output function is $\nu_{i+1} : \mathcal{D} \to \mathbb{R}^{d'}$, defined on the same domain $\mathcal{D}$ and co-domain $\mathbb{R}^{d'}$. This is one of the main reasons for the high memory demands of previous implementations of neural operators, which occurs primarily from the global integration step. In this work, we avoid this problem, since as we move to the innermost layers of the neural operator, we contract the domain of each layer while increasing the dimension of co-domains. As a result we sequentially encode the input functions into functions with smaller domain, learning compact representation as well as resulting in integral operations in much smaller domains.

## 3    A U-shaped Neural Operator (*U-NO*)

In this section, we introduce the *U-NO* architecture, which is composed of several standard elements from prior works on Neural Operators, along with U-shaped additions tailored to the structure of function spaces (and maps between them). Given an input function $a : \mathcal{D}_\mathcal{A} \to \mathbb{R}^{d_\mathcal{A}}$, we first apply a point-wise operator $P$ to $a$ and compute $v_0 : \mathcal{D}_\mathcal{A} \to \mathbb{R}^{d_0}$. The point-wise operator $P$ is parameterized with a function $P_\theta : \mathbb{R}^{d_\mathcal{A}} \to \mathbb{R}^{d_0}$ and acts as

$$v_0(x) = P_\theta(a(x)), \ \forall x \in \mathcal{D}_0$$

where $\mathcal{D}_0 = \mathcal{D}_\mathcal{A}$. The function $P_\theta$ can be matrix or more generally a deep neural network. For the purpose of this paper, we take $d_0 \gg d_\mathcal{A}$, making $P$ a lifting operator.

Then a sequence of $L_1$ non-linear integral operators is applied to $v_0$,

$$\mathcal{G}_i : \{v_i : \mathcal{D}_i \to \mathbb{R}^{d_{v_i}}\} \to \{v_{i+1} : \mathcal{D}_{i+1} \to \mathbb{R}^{d_{v_i+1}}\} \quad \text{for } i \in \{0, 1, 2 \ldots L_1\},$$

where for each $i$, $\mathcal{D}_i \subset \mathbb{R}^d$ is a measurable set accompanied by a measure $\mu_i$. Application of this sequence of operators results in a sequence of intermediate functions $\{v_i : \mathcal{D}_i \to \mathbb{R}^{d_{v_i}}\}_{i=1}^{L_1+1}$ in the encoding part of *U-NO*.

In this study, without loss of generality, we choose Lebesgue measure $\mu$ for $\mu_i$'s. Over this first $L_1$ layers, i.e., the encoding half of *U-NO*, we map the input function to a set of vector-valued functions with increasingly contracted domain and higher dimensional co-domain, i.e. for each $i$, we have $\mu(D_i) \geq \mu(D_{i+1})$ and $d_{v_{i+1}} \geq d_{v_i}$.

Then, given $v_{L_1+1}$, another sequence of $L_2$ non-linear integral operator layers (i.e. the decoder), is applied. In this stage, the operators gradually expand the domain size and decrease the co-domain dimension to ultimately match the domain of the terget function, i.e., for each $i$ in these layers, we have $\mu(D_{i+1}) \geq \mu(D_i)$ and $d_{v_i} \geq d_{v_{i+1}}$.

To include the skip connections from the encoder part from the encoder part to the decoder in (see Fig. 1), the operator $G_{L_1+i}$ in the decoder part, takes a vector-wise concatenation of both $v_{L_1+i}$ and $v_{L_1-i}$ as its input. The vector-wise concatenation of $v_{L_1+i}$ and $v_{L_1-i}$ is a function $v'_{L_1+i}$ where,

$$v'_{L_1+i} : \mathcal{D}_{L_1+i} \to \mathbb{R}^{d_{(L_1-i)}+d_{(L_1+i)}} \text{ and } v'_{L_1+i}(x) = \left[v_{L_1-i}(x)^\top, v_{L_1+i}(x)^\top\right]^\top, \ \forall x \in \mathcal{D}_{L_1+i}.$$

$v'_{L_1+i}$ constitutes the input to the operator $G_{L_1+i}$, and we compute the output function of the next layer as,

$$v_{(L_1+i)+1} = \mathcal{G}_{i+L_1} v'_{i+L_1}.$$

Here, for simplicity, we assumed $D_{L_1+i} = D_{L_1-i}$[1].

---

[1]More generally, concatenation can be defined with a map between the domains $m : D_{L_1+i} \to D_{L_1-i}$ as $v'_{i+L_1}(x) = \left[v_{L_1-i}\big(m(x)\big), v_{L_1+i}\big(x\big)\right]^\top$

Computing $v_{L+1}$ for the layer $L = L_1 + L_2$, we conclude *U-NO* architecture with another point-wise operator $Q$ kernelized with the $Q_\theta : \mathbb{R}^{d_{L+1}} \to \mathbb{R}^{d_\mathcal{U}}$ such that for $u = Qv_{L+1}$ we have

$$u(x) = Q_\theta(v_{L+1}(x)), \forall x \in \mathcal{D}_{L+1}$$

where $D_{L+1} = D_\mathcal{U}$. In this paper we choose $d_{L+1} \gg d_\mathcal{U}$ such that $Q$ is a projection operator. In the next section, we instantiate the *U-NO* architecture for two benchmark operator learning tasks.

## 4 Empirical Studies

In this section, we describe the problem settings, implementations, and empirical results.

### 4.1 Darcy Flow Equation

Darcy's law is the simplest model for fluid flow in a porous medium. The 2-d Darcy Flow can be described as a second-order linear elliptic equation with a Dirichlet boundary condition in the following form,

$$-\nabla \cdot (a(x)\nabla u(x)) = f(x) \qquad\qquad x \in D$$
$$u(x) = 0 \qquad\qquad x \in \partial D$$

where $a \in \mathcal{A} \subseteq L^\infty(D; \mathbb{R}_+)$ represents the diffusion coefficient, $u \in \mathcal{U} \subseteq H_0^1(D; \mathbb{R})$ is the velocity function, and $f \in \mathcal{F} = H^{-1}(D; \mathbb{R})$ is the forcing function. $H$ represents Sobolev space with $p = 2$.

We take $D = (0,1)^2$ and aim to learn the solution operator $G^\dagger$, which maps a diffusion coefficient function $a$ to the solution $u$, i.e., $u = G^\dagger(a)$. Please note that due to the Dirichlet boundary condition, we have the value of the solution $u$ along the boundary $\partial D$.

For dataset preparation, we define $\mu$ as a pushforward of the Gaussian measure by operator $\psi_\#$ as $\mu = \psi_\# \mathcal{N}(0, (-\Delta + 9I)^{-2})$ as a probability measure with zero Neumann boundary conditions on the Laplacian, i.e., the Laplacian is 0 along the boundary. The $\mathcal{N}(0, (-\Delta + 9I)^{-2})$ describes a Gaussian measure with covariance operator $(-\Delta + 9I)^{-2}$. The $\psi(x)$ is defined as

$$\psi(x) = \begin{cases} 3 & \text{if } x < 0 \\ 12 & \text{if } x \geq 0 \end{cases}$$

The diffusion coefficients $a(x)$ are generated according to $a \sim \mu$ and we fix $f(x) = 1$. The solutions are obtained by using a second-order finite difference method (Larsson & Thomée, 2003) on a uniform $421 \times 421$ grid over $(0,1)^2$. And solutions of any other resolution are down-sampled from the high-resolution data. This setup is the same as the benchmark setup in the prior works.

### 4.2 Navier-Stokes Equation

The Navier-Stokes equations describe the motion of fluids taking into account the effects of viscosity and external forces. Among the different formulations, we consider the vorticity-streamfunction formulation of the 2-d Navier-Stokes equations for a viscous and incompressible fluid on the unit torus ($\mathbb{T}^2$), which can be described as,

$$\partial_t w(x,t) + \nabla^\perp \psi \cdot \nabla w(x,t) = \nu \Delta w(x,t) + g(x), \qquad x \in \mathbb{T}^2, t \in (0,\infty),$$
$$-\Delta \psi = \omega, \qquad x \in \mathbb{T}^2, t \in (0,\infty),$$
$$w(x,0) = w_0(x), \qquad x \in \mathbb{T}^2,$$

Here $u \in \mathbb{R}_+ \times \mathbb{T}^2 \to \mathbb{R}^2$ is the velocity field. And $w$ is the out-of-plane component of the vorticity field $\nabla \times u$ (curl of $u$). Since we are considering 2-D flow, the velocity $u$ can be written by splitting it into components as

$$\left[u_1(x_1, x_2), u_2(x_1, x_2), 0\right] \quad \forall (x_1, x_2) \in \mathbb{T}^2$$

And it follows that $\nabla \times u = (0, 0, w)$.

The stream function $\psi$ is related to velocity by $u = \nabla^{\perp} \psi$, where $\nabla^{\perp}$ is the skew gradient operator, which is defined as

$$\nabla^{\perp} \psi = (-\frac{\partial \psi}{\partial y}, \frac{\partial \psi}{\partial x})$$

The function $g$ is defined as the out-of-plane component of the curl of the forcing function, $f$, i.e., $(0, 0, g) := (\Delta \times f)$ and $g \in \mathbb{T}^2 \to \mathbb{R}$, and $\nu \in \mathbb{R}_+$ is the viscosity coefficient.

We generated the initial vorticity $w_0$ from Gaussian measure as $(w_0 \sim \mathcal{N}(0, 7^{1.5}(-\Delta + 49I)^{-2.5})$ with periodic boundary condition with the constant mean function 0 and covariance $7^{1.5}(-\Delta + 49I)^{-2.5}$. We fix $g$ as

$$g(x_1, x_2) = 0.1\big(\sin(2\pi(x_1 + x_2)) + \cos(2\pi(x_1 + x_2)))\big), \ \forall(x_1, x_2) \in \mathbb{T}^2$$

Following (Kovachki et al., 2021b), the equation is solved using the pseudo-spectral split-step method. The forcing terms are advanced using Heun's method (an improved Euler method). And the viscous terms are advanced using a Crank–Nicolson update with a time-step of $10^{-4}$. Crank–Nicolson update is a second-order implicit method in time (Larsson & Thomée, 2003). We record the solution every $t = 1$ time unit. The data is generated on a uniform $256 \times 256$ grid and are downsampled to $64 \times 64$ for the low-resolution training.

For this time-dependent problem, we aim to learn an operator that maps the vorticity field covering a time interval spanning $[0, T_{in}]$, into the vorticity field for a later time interval, $(T_{in}, T]$,

$$G^{\dagger} : C([0, T_{in}]; H_{\text{per}}^r(\mathbb{T}^2; \mathbb{R})) \to C((T_{in}, T]; H_{\text{per}}^r(\mathbb{T}^2; \mathbb{R}))$$

where $H_{\text{per}}^r$ is the periodic Sobolev space $H^r$ with constant $r \geq 0$. In terms of vorticity, the operator is defined by $w|_{\mathbb{T}^2 \times [0, T_{in}]} \to w|_{\mathbb{T}^2 \times (T_{in}, T]}$.

## 4.3 Model Implementation

We present the specific implementation of the models used for our experiments discussed in the remainder of this section. For the internal integral operators, we adopt the approach of Li et al. (2020a), in which, evoking convolution theorem, the integral kernel (convolution) operators are computed by the multiplication of Fourier transform of kernel convolution operators and the input function in the Fourier domain. Such operation is approximately computed using fast Fourier transform, providing computational benefit compared to the other Nyström integral approximation methods developed in neural operator literature, and additional performance benefits results from computing the kernels directly in the Fourier domain, evoking the spectral convergence (Canuto et al., 2007). Therefore, for a given function $v_i$, i.e., the input to the $G_i$, we have,

$$G_i v_i(x) = \sigma\left(\mathcal{F}^{-1}\big(R_i \cdot \mathcal{F}(v_i)\big)(x) + W_i v_i(x)\right)$$

Here $\mathcal{F}$ and $\mathcal{F}^{-1}$ are the Fourier and Inverse Fourier transforms, respectively. $R_i : \mathbb{Z}^d \to \mathbb{C}^{d_{i+1} \times d_i}$, and for each Fourier mode $k \in \mathbb{Z}^d$, it is the Fourier transform of the periodic convolution function in the integral convolution operator. We directly parameterize the matrix valued functions $R_i$ on $\mathbb{Z}^d$ in the Fourier domain and learn it from data. For each layer $i$, we also truncate the Fourier series at a preset number of modes,

$$k_{max}^i = |Z_{k_{max}}^i| = |k \in \mathbb{Z}^d : k_j \leq k_{max,j}^i \text{ for } j \in \{1, \dots d\}|.$$

Thus, $R_i$ is implicitly a complex valued ($k_{max}^i \times d_{v_{i+1}} \times d_{v_i}$) tensor. Assuming the discretization of the domain is regular, this non-linear operator is implemented efficiently with the fast Fourier transform. Finally, we define the point-wise residual operation $W_i v_i$ where

$$W_i v_i(x) = W_i v_i\left(s(x)\right)$$

with $x \in \mathcal{D}_{i+1}$ and $s : \mathcal{D}_{i+1} \to \mathcal{D}_i$ is a fixed homeomorphism between $\mathcal{D}_{i+1}$ and $\mathcal{D}_i$.

In the empirical study, we need two classes of operators. The first one is an operator setting where the input and output functions are defined on 2D spatial domain. And in the second setting, the input and output are defined on 3D spatio temporal setting. In the following, we describe the architecture of *U-NO* for both of these settings.

**Operator learning on functions defined on 2D spatial domains.**   For mapping between functions defined on $2D$ spatial domain, the solution operator is of form $G^\dagger : \{a : (0,1)^2 \to \mathbb{R}^{d_\mathcal{A}}\} \to \{u : (0,1)^2 \to \mathbb{R}^{d_\mathcal{U}}\}$. The *U-NO* architecture consists of a series of seven non-linear integral operators, $\{G_i\}_{i=0}^7$, that are placed between lifting and projection operators. Each layer performs a 2D integral operation in the spatial domain and contract (or expand) the spatial domain (see Fig. 1).

The first non-linear operator, $G_0$, contracts the domain by a factor of $\frac{3}{4}$ uniformly along each dimension, while increasing the dimension of the co-domain by a factor of $\frac{3}{2}$,

$$G_0 : \{v_0 : (0,1)^2 \to \mathbb{R}^{d_{v_0}}\} \to \{v_1 : \left(0, 3/4\right)^2 \to \mathbb{R}^{\lceil \frac{3}{2} d_{v_0} \rceil}\}.$$

Similarly, $G_1$ and $G_2$ sequentially contract the domain by a factor of $\frac{2}{3}$ and $\frac{1}{2}$, respectively, while doubling the dimension of the co-domain. $G_3$ is a regular Fourier operator layer and thus the domain or co-domain of its input and out function spaces stay the same. The last three Fourier operators, $\{G_i\}_{i=4}^6$, expand the domain and decrease the co-domain dimension, restoring the domain and co-domain to that of the input to $G_0$.

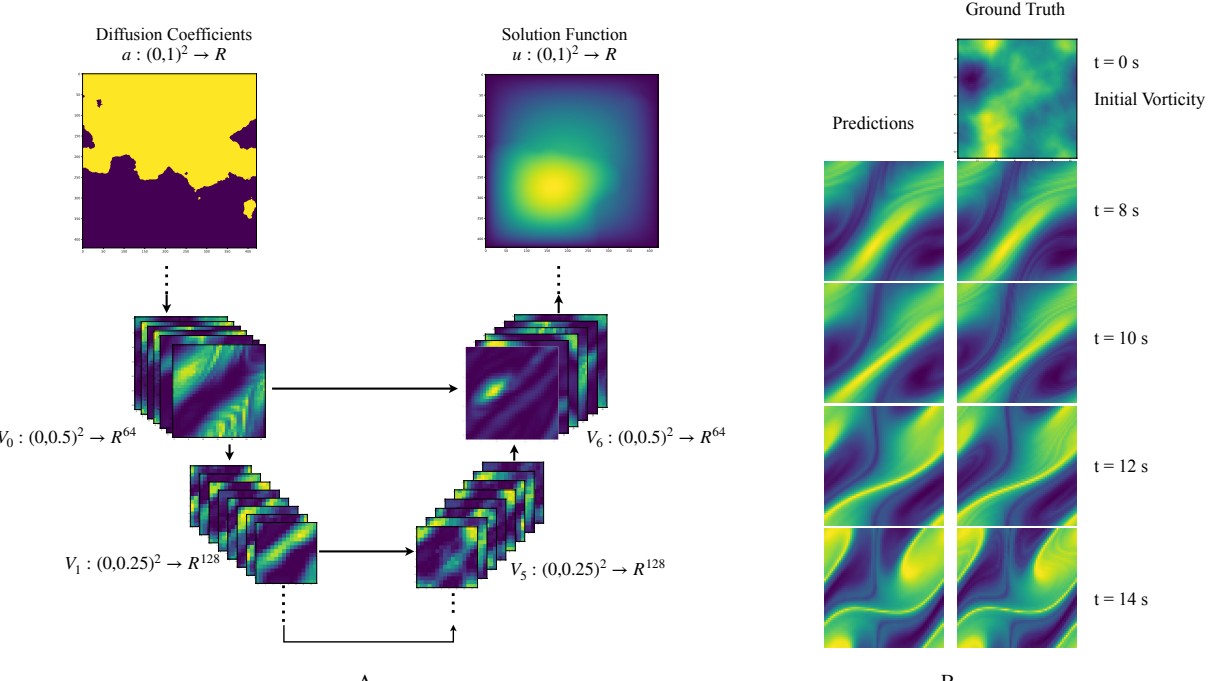

Figure 2: (A) An illustration of aggressive contraction and re-expansion of the domain (and vice versa for the co-domains) by the variant of *U-NO* with a factor of $\frac{1}{2}$ on an instance of Darcy flow equation. (B) Vorticity field generated by the *U-NO* variant as a solution to an instance of the two-dimensional Navier-Stokes equation with viscosity $10^{-6}$.

We further propose $U\text{-}NO^\dagger$ which follows a more aggressive factor of $\frac{1}{2}$ (shown in Fig. 2A) while contracting (or expanding) the domain in each of the integral operators. One of the reasons for this choice is that such a selection of scaling factors is more memory efficient than the initial *U-NO* architecture.

The projection operator $Q$ and lifting operator $P$ are implemented as fully-connected neural networks with lifting dimension $d_0 = 32$. And we set skip connections from the first three integral operator $(G_0, G_1, G_2)$ respectively to the last three $(G_6, G_5, G_4)$.

For time dependent problems, where we need to map between input functions $a$ of an initial time interval $[0, T_{in}]$, to the solution functions $u$ of later time interval $(T_{in}, T]$, i.e., to learn an operator of form,

$$G^\dagger : C\big([0, T_{in}], \{a : (0,1)^2 \to \mathbb{R}^{d_A}\}\big) \to C\big(\{(T_{in}, T], u : (0,1)^2 \to \mathbb{R}^{d_U}\}\big)$$

we use an auto regressive model for *U-NO* and *U-NO*$^\dagger$. Here recurrently compose the neural operator in time to produce solution functions for the time interval $(T_{in}, T]$.

**Operator learning on functions defined on 3D spatio-temporal domains.** Separately, we also use a *U-NO* model with non-linear operators that performs the integral operation in space and time ($3D$), which directly maps the input functions $a$ of interval $[0, T_{in}]$ to the later time interval $(T_{in}, T]$, without any recurrent composition in time. As this model does not require recurrent composition in time, it is fast during both training and inference. We redefine both $a$ and $u$ on $3D$ spatio-temporal domain and learn the the operator

$$G^\dagger : \{a : (0,1)^2 \times [0, T_{in}] \to \mathbb{R}^{d_A}\} \to \{u : (0,1)^2 \times (T_{in}, T] \to \mathbb{R}^{d_U}\}$$

The non-linear operators constructing *U-NO*, $\{G_i\}_{i=0}^L$, are defined as,

$$G_i : \big\{v_i : (0, \alpha_i)^2 \times \mathcal{T}_i \to \mathbb{R}^{d_{v_0}}\big\} \to \big\{v_{i+1} : \big(0, c_i^s \alpha_i\big)^2 \times (T_{in}, T_{in} + c_i^t T_{in}] \to \mathbb{R}^{c_i^c d_{v_0}}\big\}.$$

Here, $(0, \alpha_i)^2 \times \mathcal{T}_i$ is the domain of the function $v_i$. And $c_i^s, c_i^t$ and $c_i^c$ are respectively the expansion (or contraction) factors for spatial domain, temporal domain, and co-domain for $i$'th operator. Note that $\mathcal{T}_0 = [0, T_{in}]$, $(0, \alpha_0) = (0, \alpha_{L+1}) = (0, 1)$, and $\mathcal{T}_{L+1} = (T_{in}, T]$.

Here we also first contract and then expand (i.e., $c_i^s \leq 1$ in the encoding part and $c_i^s \geq 1$ in the decoding part) the spatial domain following the *U-NO* performing $2D$ integral operation. And we set $c_i^t \geq 1$ for all operators $G_i$ if the output time interval is larger than input time interval i.e. $T_{in} < T_{out} - T_{in}$ i.e., we gradually increase the time domain of the input function to match the output. Here the projection operator $Q$ and lifting operator $P$ are also implemented as fully connected neural network with $d_0 = 8$ for the lifting operator $P$.

For the experiments, we use Adam optimizer (Kingma & Ba, 2014) and the initial learning rate is scaled down by a factor of 0.5 every 100 epochs. As the non-linearity, we have used the GELU (Hendrycks & Gimpel, 2016) activation function. We use the architecture and implementation of *FNO* provided in the original work Li et al. (2020a). All the computations are carried on a single Nvidia GPU with 24GB memory. For each experiment, the data is divided into train, test, and validation sets. And the weight of the best-performing model on the validation set is used for evaluation. Each experiment is repeated 3 times and the mean is reported.

In the following, we empirically study the performance of these models on Darcy Flow and Navier-Stokes Equations.

### 4.4 Results on Darcy Flow Equation

From the dataset of $N = 2000$ simulations of the Darcy flow equation, we set aside 250 simulations for testing and use the rest for training and validation. We use *U-NO*$^\dagger$ with 9 layers including the integral, lifting, and projection operators . We train for 700 epochs and save the best performing model on the validation set for evaluation. The performances of *U-NO*$^\dagger$ and *FNO* on several different grid resolutions are shown in Table. 1. The number of modes used in each integral operator stays unchanged throughout different resolutions. We demonstrate that *U-NO*$^\dagger$ achieve lower relative error on every resolution with **27%** improvement on high resolution ($421 \times 421$) simulations. The U-shaped structure exploits the problem structure and gradually encodes the input function to functions with smaller domains, which enables *U-NO*$^\dagger$ to have efficient deep architectures, modeling complex non-linear mapping between the input and solution function spaces. We

also notice that *U-NO*$^\dagger$ requires **23%** less training memory while having **7.5** times more parameters. With vanilla FNO, designing such over paramterized deep operator to learn complex maps is impracticable due to high memory requirements.

Table 1: Benchmarks on Darcy Flow. The average relative error in percentage is reported with error bars are indicated in the superscript. The average back-propagation memory requirements for a single training instance is reported over different resolutions are reported. *U-NO*$^\dagger$ performs better than *FNO* for every resolution, while requiring 23% lower training memory which allows efficient training of 7.5 times more parameters

| Model | Mem. Req. (MB) | # Parameter $\times 10^6$ | Resolution $s \times s$ | | | |
|---|---|---|---|---|---|---|
| | | | s = 421 | s = 211 | s = 141 | s = 85 |
| *U-NO*$^\dagger$ | **166** | 8.2 | $0.57^{\pm 1.4e-2}$ | $0.58^{\pm 0.4e-2}$ | $0.60^{\pm 0.5e-2}$ | $0.73^{\pm 0.1e-2}$ |
| *FNO* | 214 | 1.1 | $0.78^{\pm 4.0e-2}$ | $0.84^{\pm 5.5e-2}$ | $0.84^{\pm 1.1e-2}$ | $0.87^{\pm 3.0e-2}$ |

Table 2: Zero-shot super resolution result on Darcy Flow. Neural operators trained on lower spatial resolution data is directly tested on higher resolution with no further training. We observe *U-NO*$^\dagger$ archives lower percentage relative error rate.

| | Train \ Test | s = 141 | s = 211 | s = 421 |
|---|---|---|---|---|
| *U-NO*$^\dagger$ | s=85 | 4.7 | 6.2 | 8.3 |
| | s=141 | - | 2.6 | 6.3 |
| | s=211 | - | - | 4.5 |
| *FNO* | s=85 | 7.5 | 14.1 | 23.9 |
| | s=141 | - | 4.3 | 13.1 |
| | s=211 | - | - | 9.6 |

### 4.5 Results on Navier-Stokes Equation

For the auto-regressive model, we follow the architecture for *U-NO* and *U-NO*$^\dagger$ described in Sec. 4.3. For *U-NO* performing spatio-temporal ($3D$) integral operation, we use 7 stacked non-linear integral operator following the same spatial contracting and expansion of 2D spatial *U-NO*. Additionally, as the positional embedding for the domain (unit torus $\mathbb{T}^2$) we used the Clifford torus (Appendix A.1) which embeds the domain into Euclidean space. It is important to note that, for spatio-temporal *FNO* setting, the domain of the input function is extended to match the solution function as it maps between the function spaces with same domains. Capable of mapping between function spaces with different domains, no such modification is required for *U-NO*.

For each experiment, 10% of the total number of simulations $N$ are set aside for testing, and the rest is used for training and validation. We experiment with the viscosity as $\nu \in \{1e-3, 1e-4, 1e-5, 1e-6\}$, adjusting the final time $T$ as the flow becomes more chaotic with smaller viscosities.

Table 3 demonstrates the results of the Navier-Stokes experiment. We observe that *U-NO* achieves the best performance, with nearly a **50%** reduction in the relative error over *FNO* in some experimental settings; it even achieves nearly a **1.76%** relative error on a problem with a Reynolds number[2] of $2 \times 10^4$ (Fig. 2B). In each case, the *U-NO*$^\dagger$ with aggressive contraction ( and expansion) factor, performs significantly better than *FNO*–while requiring less memory. Also for neural operators performing $3D$ integral operation, *U-NO* achieved **37%** lower relative error while requiring **50%** less memory on average with **19** times more parameters. It is important to note that designing an *FNO* architecture matching the parameter count of

---

[2]Reynolds number is estimated as $Re = \frac{\sqrt{0.1}}{\nu(2\pi)^{(3/2)}}$ (Chandler & Kerswell, 2013)

Table 3: Benchmarks on Navier-Stokes (Relative Error (%)). For the models performing a 2D integral operator with recurrent structure in time, back-propagation memory requirement per time step is reported. *U-NO* yields superior performance compared with *U-NO†* and *FNO*, while *U-NO†* is the most memory efficient model. In the 3D integration setting, *U-NO* provides significant performance improvement with almost three times less memory requirement.

| | model | Avg. Mem. Req. (MB) | # Parameters ($\times 10^6$) | $\nu = 1e{-}3$ $T = 50s$ $T_{in} = 10s$ $N = 5000$ | $\nu = 1e{-}4$ $T = 30s$ $T_{in} = 10s$ $N = 11000$ | $\nu = 1e{-}5$ $T = 20s$ $T_{in} = 10s$ $N = 11000$ | $\nu = 1e{-}6$ $T = 15s$ $T_{in} = 6s$ $N = 11000$ |
|---|---|---|---|---|---|---|---|
| | *U-NO* | 16 | 15.3 | $0.28^{\pm 2.1e{-}2}$ | $3.44^{\pm 6.3e{-}2}$ | $2.94^{\pm 4.9e{-}2}$ | $1.76^{\pm 1.4e{-}2}$ |
| 2D | *U-NO†* | **11** | 6.7 | $0.35^{\pm 2.0e{-}2}$ | $3.56^{\pm 4.3e{-}2}$ | $3.60^{\pm 1.2e{-}2}$ | $2.2^{\pm 1.5e{-}2}$ |
| | *FNO* | 13 | 1.3 | $0.58^{\pm 1.4e{-}2}$ | $5.57^{\pm 7.7e{-}2}$ | $5.12^{\pm 0.7e{-}2}$ | $3.33^{\pm 0.7e{-}2}$ |
| 3D | *U-NO* | **108** | 24.0 | $0.31^{\pm 3.9e{-}2}$ | $5.59^{\pm 2.9e{-}1}$ | $7.03^{\pm 4.6e{-}2}$ | $5.10^{\pm 1.0e{-}1}$ |
| | *FNO* | 216 | 1.3 | $0.68^{\pm 0.8e{-}2}$ | $9.60^{\pm 5.4e{-}2}$ | $8.67^{\pm 1.2e{-}1}$ | $7.35^{\pm 6.3e{-}1}$ |

*U-NO* is impractical due to high computational requirements (see Fig. 3). *FNO* architecture with similar parameter counts demands enormous computational resources with multiple GPUs and would require a long training time due to the smaller batch size. Also unlike *FNO*, *U-NO* is able to perform zero-shot super-resolution in both time and spacial domain (Appendix A.5).

Table 4: Relative Error (%) achieved by 2D *U-NO* and 3D *U-NO* trained and evaluated on high resolution ($256 \times 256$) Simulations of Navier-Stokes. Even with highly constrained neural operator architecture and smaller training data, the *U-NO* architecture can achieve comparable performance.

| Models *U-NO* | Memory Requirement (MB) | $\nu = 1e{-}3$ $N = 2400$ | $\nu = 1e{-}4$ $N = 6000$ | $\nu = 1e{-}5$ $N = 6000$ | $\nu = 1e{-}6$ $N = 6000$ |
|---|---|---|---|---|---|
| 2D *U-NO* | 86 | 0.51 | 5.9 | 7.0 | 4.4 |
| 3D *U-NO* | 850 | 0.83 | 8.3 | 11.2 | 8.2 |

Due to U-shaped encoding decoding structure–which allow us to train *U-NO* on very high resolution ($256 \times 256$) data on a single GPU with a $24GB$ memory. To accommodate training with high-resolution simulations of the Navier-Stocks equation, we have adjusted the contraction (or expansion) factors. The Operators in the encoder part of *U-NO* follow a contraction ratio of $\frac{1}{4}$ and $\frac{1}{2}$ for the spatial domain. But the co-domain dimension is increased only by a factor of 2. And the operators in the decoder part follow the reciprocals of these ratios. We have also used a smaller lifting dimension for the lifting operator $P$. Even with such a compact architecture and lower training data, *U-NO* achieves lower error rate on high resolution data (Table 4). These gains are attributed to the deeper architecture, skip connections, and encoding/decoding components of *U-NO*.

## 4.6 Remarks on Memory Usage

Compared to *FNO*, the proposed *U-NO* architectures consume less memory during training and testing. Due to the gradual encoding of input function by contracting the domain, the *U-NO* allow for deeper architectures with a greater number of stacked non-linear operators and higher resolution training data (Appendix A.4). For *U-NO*, an increase in depth, does not significantly raise the memory requirement. The training memory requirement for 3D spatio-temporal *U-NO* and *FNO* is reported in Fig. 3A.

We can observe a linear increase in memory requirement for *FNO* with the increase of depth. This makes *FNO* s unsuitable for designing and training very deep architectures as the memory requirement keep increasing

rapidly with the increase of the depth. On the other hand, due to the repeated scaling of the domain, we do not observe any significant increase in the memory requirement for *U-NO*. This make the *U-NO* a suitable architecture for designing deep neural operators. The improvement of the performance with the increase of

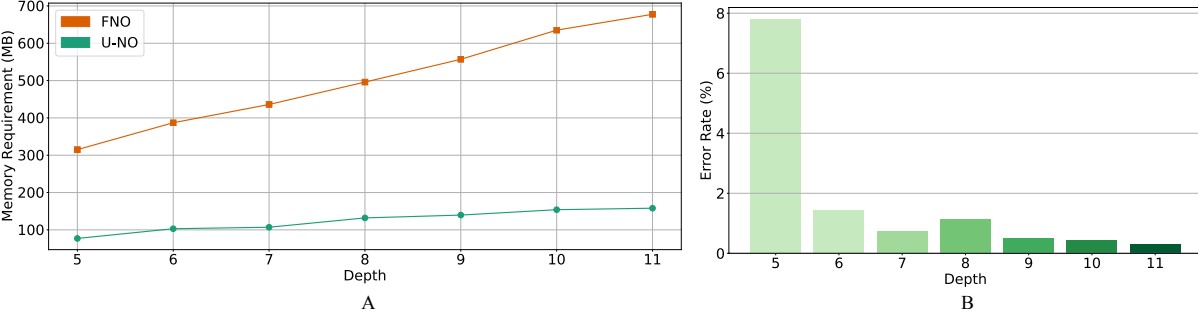

Figure 3: (A) Training memory requirements (in MB) for the 3D spatio-temporal problem of Navier-Stokes equation ($\nu = 1e-3$). For different depth only the number of stacked non-linear operators is varied. For deeper models, the additive memory requirement of *U-NO* is negligible compared to *FNO* model. For addition of 7 more integral layer memory requirement only increased by only **80MB** (vs $\sim$ **400MB** for *FNO*). (B) Relative error in percentage on 3D spatio-temporal Navier-Stokes equation ($\nu = 1e-3$) for different depth (average over three repeated experiment is reported). We can observe a gradual decrease in the relative error with the increase of depth.

depth is shown in Fig. 3B. This is expected as highly parameterized deep architectures allow for effective approximation of highly complex non-linear operators. *U-NO* style architectures are more favorable for learning complex maps between function spaces.

## 5 Conclusion

In this paper, we propose *U-NO*, a new neural operator architecture with a multitude of advantages as compared with prior works. Our approach is inspired by *Unet*, a neural network architecture that is highly successful at learning maps between finite-dimensional spaces with similar structures, e.g., image to image translation. *U-NO* possesses a similar set of benefits to *Unet*, including data efficiency, robustness to hyperparameter tuning (Appendix A.2), flexible training, memory efficiency, convergence (Appendix A.6), and non-vanishing gradients.This approach, while advancing neural operator research, significantly improves the performance and learning paradigm, still carries the fundamental and inherent memory usage limitation of integral operators. Nevertheless, *U-NO* allows for much deeper models incorporating the problem structures and provides highly parameterized neural operators for learning maps between function spaces. *U-NO*s are very easy to tune, possess faster convergence to desired accuracies, and achieve superior performance with minimal tuning.

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

# A    Appendix

## A.1    Positional Embedding For Unit Torus

A unit torus $\mathbb{T}^2$ is homeomorphic to the Cartesian product of two unit circles: $S^1 \times S^1$. We can also consider the cartesian product of the embedding of the circle. which produces Clifford torus. The Clifford torus can be defined as

$$\frac{1}{2}S^1 \times \frac{1}{2}S^1 = \left\{ \frac{1}{2}(sin(\theta), cos(\theta), sin(\phi)cos(\phi)) \mid 0 \leq \theta \leq 2\pi, 0 \leq \phi \leq 2\pi \right\}$$

## A.2    Sensitivity to Hyper-parameter Selection

Learning Rate

| | Error Rate (Training Set) | | | | | | Error Rate (Test Set) | | | | | | Generalization Gap | | | | |
|---|---|---|---|---|---|---|---|---|---|---|---|---|---|---|---|---|---|
| Depth | 0.01 | 0.005 | 0.001 | 0.0005 | 0.0001 | | 0.01 | 0.005 | 0.001 | 0.0005 | 0.0001 | | 0.01 | 0.005 | 0.001 | 0.0005 | 0.0001 |
| 7 | 0.4435 | 0.4188 | 0.3656 | 0.483 | 1.2261 | 7 | 0.8129 | 0.8209 | 0.7625 | 0.9061 | 1.8448 | 7 | 0.3694 | 0.4021 | 0.3969 | 0.4231 | 0.6187 |
| 9 | 0.4435 | 0.4188 | 0.3656 | 0.483 | 1.2261 | 9 | 0.723 | 0.5907 | 0.5539 | 0.6229 | 1.622 | 9 | 0.2795 | 0.1719 | 0.1882 | 0.1399 | 0.396 |
| 11 | 0.4276 | 0.3114 | 0.2503 | 0.2484 | 0.6188 | 11 | 0.7094 | 0.6339 | 0.6104 | 0.6402 | 1.6007 | 11 | 0.2818 | 0.3225 | 0.36 | 0.3918 | 0.9819 |
| 13 | 0.4896 | 0.2843 | 0.2249 | 0.1994 | 0.5106 | 13 | 0.8946 | 0.7012 | 0.5783 | 0.5964 | 2.3528 | 13 | 0.405 | 0.4169 | 0.3534 | 0.3971 | 1.8421 |
| 15 | 0.3861 | 0.2709 | 0.2339 | 0.2573 | 0.449 | 15 | 0.8195 | 0.7558 | 0.6551 | 0.7019 | 1.9204 | 15 | 0.4334 | 0.4849 | 0.4212 | 0.4446 | 1.4714 |
| 17 | 0.3882 | 0.2847 | 0.231 | 0.2761 | 0.4474 | 17 | 0.8244 | 0.7704 | 0.6586 | 0.8074 | 1.8612 | 17 | 0.4362 | 0.4858 | 0.4275 | 0.5313 | 1.4138 |

Figure 4: Result of sensitivity of $U$-$NO^\dagger$ to learning rate and number of stacked non-linear operator (depth) used. All models are trained on the dataset of the Darcy Flow equation with resolution $211 \times 211$ following the training protocol described in 4.4. Models for each of the configuration is trained three times and the average error rate (in %) is reported. We can notice that except for a very high ($\geq 0.01$) or very low ($\leq 0.0001$) learning rate, $U$-$NO^\dagger$ achieves low error rate at every other configurations (**error rate achieved by** *FNO* **is 0.85** ). We also note that at all the high-performing hyper-parameter configuration setting *U-NO* has low generalization gap.

Table 5: Number of Parameters in $U$-$NO^\dagger$ models at different depth. We observe $U$-$NO^\dagger$ architecture can efficiently learn large number of parameter (**25** times of the *FNO*) from very limited training data (only 1500 simulations) and can perform reasonably (See Table. 4)

| Depth | 7 | 9 | 11 | 13 | 15 | 17 |
|---|---|---|---|---|---|---|
| #Parameters ($\times 10^6$) | 5.3 | 8.3 | 9.6 | 18.5 | 22.8 | 27.8 |

## A.3    *FNO* with Skip Connection

Table 6: Performance of vanilla 2D *FNO* equipped with skip connection on Navier-Stokes equations described in Section 4.5 along with memory requirement during training. The result of 2D *FNO* is reported again for comparison. We notice that skip connection alone does not improve the performance of *FNO*.

| Models U-NO | Memory Requirement (MB) | $\nu = 1e{-}3$ $N = 2400$ | $\nu = 1e{-}4$ $N = 6000$ | $\nu = 1e{-}5$ $N = 6000$ | $\nu = 1e{-}6$ $N = 6000$ |
|---|---|---|---|---|---|
| *FNO* w. skip con. | 13.14 MB | 0.011 | 0.074 | 0.075 | **0.049** |
| *FNO* | **13.03 MB** | **0.009** | **0.072** | **0.074** | 0.052 |

## A.4    Spatial Memory

The memory requirements to learn the operator $w|_{(0,1)^2 \times [0,10]} \to w|_{(0,1)^2 \times (10,11]}$ during back-propagation for the Navier-Stokes problem are shown in Table 7. On average, for various tested resolutions, *U-NO* with a

contraction (and expansion) factor of $\frac{1}{2}$ requires 40% less memory than *FNO* during training. This makes *U-NO* and its variants more suitable for problems where high-resolution data are crucial, e.g., weather forecasting model (Pathak et al., 2022).

Table 7: Memory requirements (in MB) for a single training instance of the Navier-Stokes equations for different grid resolutions. All models have seven non-linear integral operator layers.

| Spatial Resolution | *FNO* | *U-NO* | *U-NO*$^\dagger$ |
|---|---|---|---|
| $64 \times 64$ | 13.0 | 16.8 | 11.3 |
| $128 \times 128$ | 76.1 | 67.3 | 45.4 |
| $256 \times 256$ | 304.5 | 269.0 | 181.5 |
| $512 \times 512$ | 1218.0 | 1076.0 | 726.0 |
| $1024 \times 1024$ | 4872.0 | 4304.0 | 2990.9 |

## A.5 Zero-Shot super resolution on 3D Spatio-Temporal Data

Table 8: Zero-shot super resolution result on 3D spatio-temporal Navier Stocks equation. The 3D *FNO* is not resolution invariant in the time domain and due the specific construction can not process data at different temporal resolution. But *U-NO* is resolution invariant both in time and spatial domain.

| | Temporal Res. Spatial Res. | fps = 1 | fps = 1.5 | fps = 2 | fps = 3 |
|---|---|---|---|---|---|
| | s=64 | 5.10 | 17.43 | 19.39 | 20.32 |
| *U-NO* | s=128 | 6.34 | 17.61 | 19.56 | 20.62 |
| | s=256 | 8.16 | 17.86 | 19.94 | 20.83 |

## A.6 Superior Convergence Rate

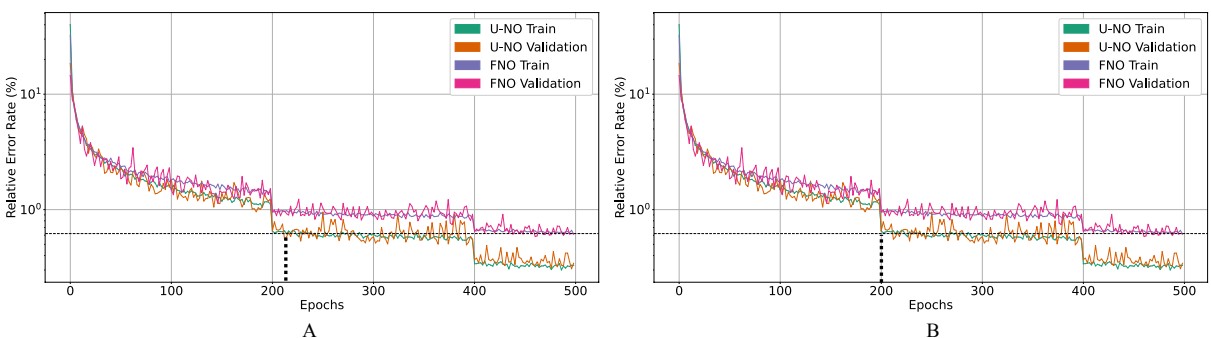

Figure 5: The training and test set error rate (in % on log scale) of *U-NO* and *FNO* for Navier-Stokes equation with viscosity $10e^{-3}$, (A) models performing $2D$ spatio-temporal convolution (B) models performing $3D$ spatial covolution. We can notice what *U-NO* converges much faster than *FNO*. The final test set error rate for *FNO* after **500** epochs is achieved only after around **200** epochs by *U-NO* and continues to improve on the error rate.

## A.7 Domain Contraction and Expansion

In this work, the domain contraction (or expansion) is performed by setting homeomorphism or mapping between the domains of the input and output function of an integral operator. Following the notations of Sec.

4.3, an integral operator can be defined as

$$[G_iV_i](x) = V_{i+1}(x) = \sigma\left(\mathcal{F}^{-1}\big(R_i \cdot \mathcal{F}(v_i)\big)(s(x)) + W_iv_i(s(x))\right)$$

Where with $x \in \mathcal{D}_{i+1}$ and $s : \mathcal{D}_{i+1} \to \mathcal{D}_i$ is a fixed homeomorphism between $D_{i+1}$ and $D_i$. For the discussed problems in this work (Darcy flow and Navier Stokes equation), the domains of the input and output functions are bounded and connected. As a result, the mapping can be trivially established by scaling operation. Lets the domain of the input and the output functions be $(a,b)^2$ and $(c,d)^2$ respectively, and the function $s : [c,d] \to [a,b]$ is a homeomorphism between them. The function $s$ has a linear form $s(x) = a + m \circ (x - c)$ with $m = (b-a) \oslash (d-c)$ where $\circ$ and $\oslash$ represents element wise multiplication and division. If $c = a = 0$, $s$ becomes a scaling operation on the domain. And the resultant output function can be efficiently computed through interpolation with appropriate factors along each dimension.

## A.8   Training Memory Requirement for 2D *U-NO* at Different Depths

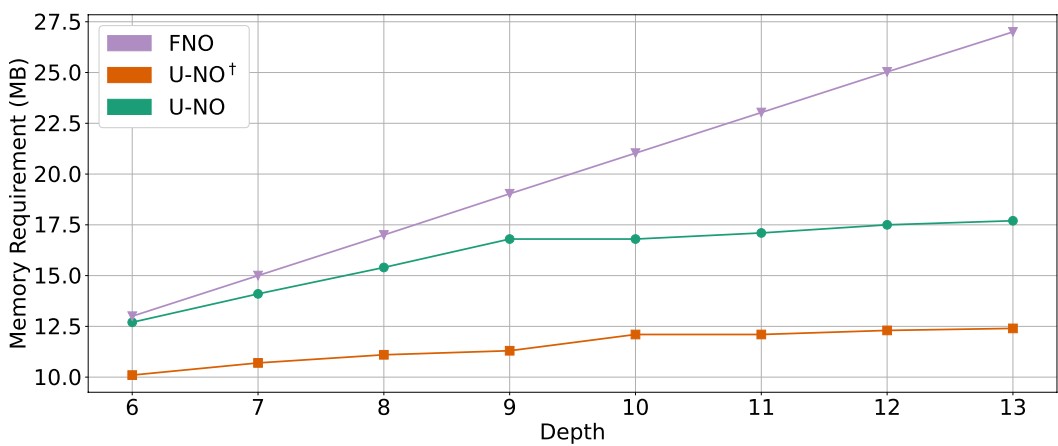

Figure 6:   Memory requirements (in MB) for a single training instance of the Navier-Stokes equation during back-propagation on input and outputs of resolution $64 \times 64$. Number of stacked non-linear operators is varied and the additive memory requirement of *U-NO* architectures for deeper models are negligible compared to *FNO* model. For *U-NO*† the addition of **7** stacked non-linear integral operator increases the memory requirement by only **2.5 MB**

## A.9   Comparison of Inference Time

As across different problem settings, the number of parameters of *U-NO* is $\sim 8 - 25$ times more than the *FNO*; it has more floating point operations (FLOPs). For this reason, the inference time of *U-NO* is more than *FNO* (see Table 9). But it is important to note that even if t *U-NO* has $\sim 8 - 25$ times more parameters, the inference time is only $\sim 1 - 4$ times more. As *U-NO* in the encoding part gradually contracts the domain of the functions, it reduces the time in the forward and inverse discrete Fourier transform.

Table 9: The running time of *U-NO* and *FNO* during inference of a single sample. For the Darcy Flow problem, we test on the resolution of $421 \times 421$, and for Navier Stocks, we use the problem setting with viscosity $10^{-3}$ with the resolution of $64 \times 64$. For Navier-Stocks 2D time to infer a single time step and for Navier-Stocks 3D time to infer the whole trajectory are reported.

| Problem | *U-NO* (sec) | *FNO* (sec) |
|---|---|---|
| Darcy Flow | 0.13 | 0.12 |
| Navier-Stcoks (2D) | 0.08 | 0.02 |
| Navier-Stocks (3D) | 0.33 | 0.09 |

