# OpenReview forum: "U-NO: U-shaped Neural Operators"
_TMLR — Accepted by TMLR_

### Review · Reviewer_L1LD · 2023-01-11

**Summary Of Contributions:**

This paper proposes a novel parameterization of Neural Operators inspired by the classical UNet architecture used to encode image-to-image functions in computer vision. This novel architecture, named UNO, compared to the prior state-of-the-art Fourier Neural Operator (FNO) is more memory efficient, trains faster, is less sensitive to hyperparameters and achieves higher generalization performances across resolutions. UNOs work by contracting the domain and expanding the co-domain of the output function in the first half of the layers, and doing the opposite in the second half of the layers. In the spirit of the classical UNets, skip connections are used to connect the representations that live in the same functional spaces at mirroring layers. Results on two standard benchmarks for Neural Operators show that UNOs exceed the performance of FNOs in all reported metrics.

**Audience:**

Yes

**Broader Impact Concerns:**

I see no clear broader impact concerns.

**Claims And Evidence:**

Yes

**Requested Changes:**

I would suggest that the authors work on addressing my weaknesses, mostly focusing on 1., and 2., in their revised version of the text. I believe that will make for a much stronger submission.

**Strengths And Weaknesses:**

**Disclaimer:** I am no expert in neural operators and do not know in depth the related literature. I am an expert in deep learning with a good mathematical knowledge in PDEs and functional analysis. I had read (Li et al. 2020) and (Kovachki et al. 2021) but I have never worked in the field. My review should therefore be read as a low-confidence review in which I mostly focused on methodology and presentation. I would differ opinions about prior work and deep technical details to the other reviewers.

## Strenghts

1. **Simple, elegant and effective technique:** The idea of using a UNet style architecture to circumvent the issues of prior FNO-style architectures is a natural decision with seemingly great results.
2. **Clear writing**: The paper is very well written and very enjoyable to read.
3. **Strong and thorough evaluations:** Methodologically, I really appreciate the effort made by the authors in comparing the performance of different versions of UNO to FNO with respect to many different metrics. The results in the Appendix nicely complement the main paper giving further insights about UNO (see for example ablation study using skip connections on FNO).
4. **Reproducibility**: The authors provide easy-to-follow code implementing and reproducing their main training loops used with UNOs on the two datasets.

## Weaknesses

1. **Unclear how the domain contraction affects memory:** It might be due to my lack of knowledge of the field, but reading the paper I do not fully understand why contracting the domain affects the memory usage of the architecture. I clearer explanation of how this is implemented in the text would be helpful for less expert readers.
2. **No details about FNO architecture used**: As I understand it, the key message of this paper is that UNO is a more powerful architecture than FNO. However, as far I can tell, FNO is not a particular architecture, but a familiy of architectures, and depending on the specific choices of number of layers and widths, the performance of an FNO would vary. In this sense, I am missing the details in the text (or the code for that matter) explaining which specific architecture was used as baseline FNO. In particular, I am interested in knowing if you can scale the parameter count of an FNO as high as a UNO, does it perform as well albeit with a much higher memory footprint?
3. **Some mathematical notation is not defined**: Things like the push-forward operator $\psi_#$, GP notation, etc are not defined in the text. I would in general make sure that every symbol that is used in the text is introduced somewhere.

---

> ### Author Response · Authors · 2023-01-14
> **Response to the Reviewer L1LD**
>
> We thank the reviewer for appreciating our work and the valuable feedback. We list the response to the weaknesses below:
>
> **1 )** For computational feasibility, we assume a finite discretization of the domain for the function and represent the function as a finite set of points. At a fixed sampling rate, doubling the domain size doubles the number of points required to represent the function. For example, at a sampling rate of 1/10, a continuous-time function over $t \in [0,1]$ is represented by 10 points vs. 5 points for a function over  $t \in [0,0.5]$. During backpropagation, the activations at the internal layers of a model are saved for gradient calculation. Memory requirement increases with the number of points used to represent the function. By contracting the domain, we progressively define functions over smaller domains, requiring fewer data points for representation at a fixed sampling rate. Thus requiring less training memory.
>
> In this work, the domain contraction (or expansion) is performed by setting homeomorphism or mapping between the domains of the input and output function of an integral operator. $$G_iV_i(x) = V_{i+1}(x) = \sigma \big( F^{ −1}( R_i · F(v_i))  (s(x)) + W_iv_i(s(x)) \big)$$ where $x \in D_{i+1}$ and $s: D_{i+1} \rightarrow D_i$ is a fixed homeomorphism between $D_{i+1}$ and $D_i$.
> For the Navier Stokes and Darcy flow problems, the domains of the input and output functions are connected and bounded. As a result, the mapping can be trivially established by scaling operation. As an example, lets the domain of the input and the output functions be (a,b) and (c,d) respectively, and the function $s:(c,d) \rightarrow (a,b)$ is a homeomorphism between them. The function s has a linear form $s(x)=a+m(x−c)$   with $m= (b-a)/(d-c)$. If $c = a = 0$, s becomes a scaling operation on the domain. And the resultant output function can be efficiently computed through interpolation.
>
> **We thank the reviewer for pointing these out. A section addressing the details of domain contraction is added in the supplementary (Appendix A.7). Also, the effect of domain contraction leading to lower training memory requirement is clarified in the main text (Introduction).**
>
> **2 )** We have used the architecture and implementation provided in the original work of Fourier Neural Operators [1]. **We have added this information in the "Empirical Studies" section.**
> If we design FNO architecture matching the parameter count of UNO, it should perform at least as good as UNO. But showing that empirically using a considerable-sized dataset is challenging due to the following reason.
> We show the training memory requirement of the FNO-3D for a single training instance in Table 3 at different depths. An FNO-3D architecture of depth 7 ( used in [1]) requires ~420MB of memory per training sample (Navier Stocks problem with viscosity 10^-3) and can be trained on a single GPU (24GB) with a maximum batch size of 4.  A very deep FNO architecture should be designed to match the parameter count of UNO, which will require enormous computational resources with multiple GPUs and a long training time due to the smaller batch size.
>
> **3 )** We thank the reviewer for indicating that. We have added explanations to the mathematical notations in the revised text.
>
> 1. Li, Z., Kovachki, N., Azizzadenesheli, K., Liu, B., Bhattacharya, K., Stuart, A. and Anandkumar, A., 2020. Fourier neural operator for parametric partial differential equations. arXiv preprint arXiv:2010.08895

---

> > ### Comment · Reviewer_L1LD · 2023-01-24
> > **Thaks for the clarifications**
> >
> > Thank you very much for clarifying these points. In my opinion, the text is now more clear and the main contribution of UNO (contraction/expansion mappings allow memory savings) is also better explained.
> >
> > As for point 2), I would suggest the authors explicitly added this discussion in their text, as i believe it is an important detail.
> >
> > In any case, from my side, I believe all my concerns have been addressed. I would like to hear from other reviewers who are deeper experts in the field, but in my opinion this is a solid contribution.

---

> > > ### Author Response · Authors · 2023-03-09
> > > **Response to the Reviewer L1LD**
> > >
> > > Thank you, reviewer L1LD for pointing this out. We have added Point 2 in the main text (section 4.5). Please let us know if you need any further clarifications.

---

> ### Comment · Reviewer_L1LD · 2023-03-13
> **Update after reading other reviews**
>
> After reading the other reviews and the comments of the other reviewers, I stand by my previous assessment and I wil strongly recommend acceptance of this work. I believe this paper clearly meets the bar for acceptance on the two acceptance criteria of TMLR as it is factually correct/methodologically precise, and provides a very actionable and clear contribution to readers interested in Neural Operators. The results of this paper are strong and reliable, and I believe the authors have clearly shown they followed proper evaluation practices to arrive to them. The paper is also very clear and easy to read. Specifically, the authors have made a nice effort in improving the accessibility of the paper to non-experts in Neural Operators.

---

### Review · Reviewer_tUW5 · 2023-01-30

**Summary Of Contributions:**

**Note:** My review will be from the perspective of someone unfamiliar with this type of field (i.e. solving partial differential equations with neural networks)

At a base case, Neural Operators are the "neural network" equivalent over functions, rather than simple tensors, where "input functions" are mapped to "output functions" via a MLP-like transformation seen in Eq (1). Unfortunately, the regular Neural Operator forces the "output function" to have the same exact domain/co-domain as the "input function". This has the consequence of large memory usage, especially since the Neural Operator needs to perform an integration over the domain.

Instead, the paper proposes a U-Net equivalent of Neural Operators, in which the operators on the "encoder" side can shrink the output function's domain, which leads to lower computation. Like the U-Net architecture, the outputs are also passed to the decoder side for concatenation.

Experiments are conducted on well-known differential equations from physics, in which the "input functions" are problem parameters or initial conditions, mapped to the final solutions. Results demonstrate better accuracy (especially with more operator layers), in addition to much lower memory usage, even when multiple depths are used.





**Audience:**

Yes

**Broader Impact Concerns:**

No borader impact concerns

**Claims And Evidence:**

Yes

**Requested Changes:**

Please address my "Weaknesses" section above, as they are the largest requested changes.


Minor Potential Typos/Questions for clarification:
1. In Equation (1), do you mean $\kappa_{i}(x,y)v_{i}(x) d \mu_{i}(y)$? If not, what is the meaning of $v_{i}$ (without the input $x$)?
2. Page 6, middle of page - an additional parenthesis is unneeded, $W_{i} v_{i}(x) = W_{i} v_{i}(s(x))$

**Strengths And Weaknesses:**

Strengths:
* In general, the paper's contributions are solid both in formulation and empirically. The paper supports its claims and evidence fairly well.
* Notation, while dense, is precise and after some re-reading, I could follow through.
* The paper demonstrates good applications for the scientific computing / physical simulation industry


Weaknesses:
* The paper can potentially relax the notation, or at least give better high-level overviews of the core idea (e.g. see my summary of contributions). I needed to read the paper multiple times until I understood the high-level strategy.
* Furthermore, because I am not a physics expert, I briefly read, but did not fully appreciate/understand the Darcy Flow Equations / Navier-Stokes Equations nor their baseline solvers - some of their details appeared "out of nowhere". These could also be explained more accessibly to the common reader.
* Please clarify: Is the main contribution that U-NO is more memory/compute-efficient (and therefore one can stack more depths to improve accuracy)? For now, it's unclear why skip-connections are needed (do they empirically help, or could they even be removed?)
* Could you explain more on the semantic meaning of the domain compression? i.e. Because it works, does it suggest that the "solution spaces" of certain differential equations lie in lower dimensions?

---

> ### Author Response · Authors · 2023-03-09
> **Response to the Reviewer tUW5**
>
> We thank reviewer tUW5 for admiring our work and giving very insightful suggestions to increase the quality of the work.
> We list our response to weaknesses below:
>
> 1. The notations introduced in the paper are necessary to generalize the u-shaped architecture for neural operators. We agree with the reviewer that a brief overview is needed. We have added a comprehensive overview of the pipeline in the introduction of the revised version (page 2).
>
> 2. We thank the reviewer for suggesting this.  We have added further clarification of physics problems and the formulation of the partial differential equations in sections 4.1 and 4.2. We also referenced the numerical solution scheme used as the baseline solver.
>
> 3. We agree with the reviewer that memory/compute-efficient architecture is the main contribution. Principally they design UNO by proposing an integral operator that can map between functions with different domains. The skip layer is necessary to pass information across different domain sizes from the encoder side to the decoder bypassing the bottleneck of the encoder layer. We clarified this point in the revised main text (page 2).
>
> 4. Please consider the following example. Let the operator $G$ is defined as $G: \set{u: (0,1) \rightarrow R} \rightarrow \set{v: (0,0.5) \rightarrow R}$. Here the operator maps the function defined on the domain (0,1) (the $u$ s) to the functions defined on the domain $(0,0.5)$ (the $v$ s). The domain of the output functions is contacted by a factor of 2.
>
> We thank the reviewer for pointing to the typos in equation 1 and page 6. We have corrected both of them in the revised text. Please let us know if any further clarification is needed.

---

> > ### Comment · Reviewer_tUW5 · 2023-03-17
> > **LGTM**
> >
> > Thanks for addressing my concerns - paper reads cleaner now.

---

### Review · Reviewer_ahuP · 2023-03-06

**Summary Of Contributions:**

This paper proposes U-shaped Neural Operator (U-NO) that combines U-Net architecture and neural operators to allow deeper models. Other advantages of U-NO, as claimed in the paper, include fast training, data efficiency, and robustness with respect to hyperparameter values. Experimental results on Darcy’s flow and Navier-Stokes are provided to justify the advantage of the proposed method.

**Audience:**

Yes

**Broader Impact Concerns:**

The paper has no ethical concern or implication that would require adding a Broader Impact Statement.

**Claims And Evidence:**

No

**Requested Changes:**

1. The proposed U-NO requires much more parameters than the baseline FNO. Can the authors discuss this more?

2. The average relative errors for Darcy Flow and Navier-Stokes given in Table 1 and 3, respectively, do not match those in the original paper [1]. Also, the average relative errors in these tables are incomprehensible. I think there is a formatting bug there.

3. A comparison of the runtime of U-NO and FNO is needed.


**Strengths And Weaknesses:**

Strong points:

1. This paper proposes a useful extension of FNO.

2. The paper is well motivated

3. The paper is well written with great-looking illustrative figures.

Weak points:

1. The contribution of the paper is limited. Replacing fully-connected neural networks and convolutional neural networks in FNO by a U-Net in U-NO is incremental.

2. There is no theoretical justification for the proposed U-NO. For example, what is the theoretical result that supports the claim that U-NO allows deeper neural operators?

3. The empirical results in the paper are not convincing.

---

> ### Author Response · Authors · 2023-03-09
> **Response to the Reviewer ahuP**
>
> We thank reviewer ahuP for the insightful reviews and suggestions. The responses are listed below.
>
> 1.**The contribution of the paper is limited. Replacing fully-connected neural networks and convolutional neural networks in FNO by a U-Net in U-NO is incremental.**
>
> In this work, we generalize the ideas behind the u-shaped architecture of U-net to neural operator settings. We provide a rigorous mathematical formulation to adapt the u-shaped architecture in the operator learning settings. The adoption of different architectural components, e.g., horizontal skip connection, residual connections, domain contraction & subsequent expansion (encoding & decoding),  in the infinite-dimensional spaces is nontrivial and requires careful redefinition. Furthermore, we empirically show that under the UNO setting, overparameterized neural operator architectures are possible, and these architectures result in improvement in the accuracy of learned models.
>
> 2.**There is no theoretical justification for the proposed U-NO. For example, what is the theoretical result that supports the claim that U-NO allows deeper neural operators?**
>
> In the paper, we argued that in UNO, as we shrink the domain, we are dealing with a smaller domain resulting in smaller memory usage (clarified in the revision, page 2).  So, given a fixed memory size, we can stack more integral operator layers and have deep architectures.
>
> 3.**The empirical results in the paper are not convincing.**
>
> In the benchmark problems, UNO outperformed the state-of-the-art FNO by a large margin while requiring less training memory. For the Darcy Flow problem, UNO achieves a 26% better error rate while requiring 23% less memory. For the Navier Stocks equation, UNO improves the relative error by 44% for 2D case. And for the 3D case, UNO improves the error rate by 37% while requiring only 50% of memory consumed by FNO 3D.  We showed imperial evidence of scalability and convergence. We believe these improvements are significant, and UNO will be useful in many practical scenarios.
>
> 4.**The proposed U-NO requires much more parameters than the baseline FNO. Can the authors discuss this more?**
>
> The proposed architecture allows us to design deep over-parameterized models. This helps UNO to attain a better result. From contemporary works, we know that the model overparameterization improves generalization performance. The U-NO architecture allows efficient training of these desired overparameterized neural operator models with smaller memory footprints.  Also, we added a discussion on why we can not achieve such models via FNO in the revised text (sec 4.5)
>
> 5.**The average relative errors for Darcy Flow and Navier-Stokes given in Table 1 and 3, respectively, do not match those in the original paper [1]. Also, the average relative errors in these tables are incomprehensible. I think there is a formatting bug there.**
>
> We report the relative error rate in percentage with the error bar over multiple trials as the superscript. Whereas the raw error rates (not the percentage) are reported in the original Fourier Neural Operator paper. Also, the number of data points used in training differs from the original paper. And unlike the original paper, all the models trained here use a validation set, and the best-performing model is saved for evaluation. Each experiment is repeated three times (with different seeds and data split), and the mean and error bar is reported. These are the probable sources of the little variations in the numerical results. While evaluating FNO, we use their official implementation.
>
> 6.**A comparison of the runtime of U-NO and FNO is needed.**
> We thank the reviewer for this important suggestion. We added a section discussing the run time of the UNO and FNO in the appendix (section A9)

---

> > ### Comment · Reviewer_ahuP · 2023-04-18
> > **Thank you for the response!**
> >
> > I would like to thank the authors for the reply. After reading the rebuttal, I still believe that the contribution of the paper is limited without enough theoretical guarantees and empirical evidences to support the advantage of the proposed method. At this moment, I am still leaning toward rejecting the paper.

---

### Decision · Action_Editors · 2023-04-27

**Recommendation:** Accept as is

**Comment:**

I would like to cite one of the reviewers here:
"After reading the other reviews and the comments of the other reviewers, I stand by my previous assessment and I will strongly recommend acceptance of this work. I believe this paper clearly meets the bar for acceptance on the two acceptance criteria of TMLR as it is factually correct/methodologically precise, and provides a very actionable and clear contribution to readers interested in Neural Operators. The results of this paper are strong and reliable, and I believe the authors have clearly shown they followed proper evaluation practices to arrive at them. The paper is also very clear and easy to read. Specifically, the authors have made a nice effort in improving the accessibility of the paper to non-experts in Neural Operators."

I fully agree with this statement and, thus, I strongly suggest accepting the paper.

**Audience:**

The paper is of interest to the TMLR's audience. First, the application of the U-shaped parameterization is interesting (e.g., a clear connection to diffusion models and its parameterization using UNet). Second, it is of great importance to the growing AI4Science (or scientific ML) community.


**Claims And Evidence:**

The paper proposes the U-shaped Neural Operator (U-NO) and its claims are the following:
1) The U-NO architecture can be adapted to the existing operator learning techniques.
2) The U-NO architecture achieves better results than the state-of-the-art FNO.
3) The U-NO architecture is faster to train and easier to tine than FNO.

The authors successfully provided evidence for all of the claims. The paper is easy to follow and after addressing most of the reviewers' comments, it reads even better. Two out of the three reviewers agree that the paper should be accepted. In my assessment, the paper is an interesting contribution and should be accepted as it is.